# Analysis of the Predictors and Consequential Factors of Emotional Exhaustion Among Social Workers: A Systematic Review

**DOI:** 10.3390/healthcare13050552

**Published:** 2025-03-04

**Authors:** Alfonso Chaves-Montero, Pilar Blanco-Miguel, Belén Ríos-Vizcaíno

**Affiliations:** 1Department of Sociology, Social Work and Public Health, Faculty of Social Work, COIDESO Research Centre, Contemporary Thought and Innovation for Social Development, University of Huelva, 21007 Huelva, Spain; alfonso.chaves@dstso.uhu.es; 2Department of Sociology, Social Work and Public Health, Faculty of Social Work, ESEIS Research Group: Social Studies and Social Intervention, University of Huelva, 21007 Huelva, Spain; 3Department of Sociology, Social Work and Public Health, Faculty of Social Work, University of Huelva, 21007 Huelva, Spain; belen.rios@dstso.uhu.es

**Keywords:** emotional exhaustion, predictors, consequential factors, social workers, cross-cutting research

## Abstract

Background: Emotional exhaustion, a central component of burnout syndrome, affects social workers due to adverse work factors such as excessive workload, work–family conflict, and a lack of social support. The job demands–resources and conservation of resources models explain how chronic stress influences these professionals. Methods: A systematic review was carried out using the recommendations of the PRISMA guidelines as a reference for the selection and identification of studies and the Joanna Briggs Institute guidelines, registering the protocol in PROSPERO. Cross-sectional studies published from 1 January 2019 to 30 September 2024 were included and five main databases―Web of Science, Scopus, PubMed, Medline, and PsycInfo—were used to search for specific studies written in English, Spanish, French, and Portuguese. Results: Of 361 initial records, 21 studies involving more than 24,000 social workers from diverse global settings were analyzed. The main risk factors identified were workload, work–family conflict, and work victimization. Resilience, self-care, and social support were highlighted as protective factors. Emotional exhaustion was associated with low job satisfaction, turnover intention, and mental health problems such as anxiety and depression. Conclusions: Emotional burnout requires specific strategies, such as work flexibility, wellness programs, and organizational support. These measures can mitigate its impact, improving work–life balance and fostering resilience.

## 1. Introduction

The International Federation of Social Workers (2014) describes social work as a discipline that combines practical and academic aspects, focused on promoting change, social development, community cohesion, and empowerment at both the individual and collective levels. Various international studies have highlighted that professionals in this area are particularly exposed to high levels of psychological stress and burnout, attributable to the demands inherent to their work. Research has identified key factors that contribute to this problem, such as unfavorable working conditions, conflicts in the roles performed, and a lack of social support [1,2,3,4].

Burnout is a set of symptoms characterized by physical and mental exhaustion, caused mainly by extreme working conditions that demand a high level of emotional commitment [5,6]. This syndrome does not arise suddenly, but develops progressively, driven by a constant increase in stress and the person’s inability to manage an excessive mental load [6,7]. There is a direct relationship between chronic stress and emotional exhaustion, as evidenced in various studies [8,9] that have analyzed in depth the environmental and personal factors that trigger or inhibit burnout. The job demands–resources model [10,11], together with its subsequent development in the job demands and resources theories [12,13], establishes that the demands of the work environment can lead to energy depletion, fatigue, and other health problems.

In a complementary manner, the conservation of resources (COR) theory [14,15] argues that a loss of personal or work resources is associated with a significant increase in stress, which could lead to burnout. In contrast, having sufficient resources, both personal and work-related, has been shown to have a protective effect and reduce stress levels [15,16].

According to [17], burnout acts as a mediator in the relationship between stressors and job satisfaction, within the framework of the stress–strain–outcome model [18]. Workplace research suggests that burnout is influenced by several variables, such as the individual’s personality, perceived professional ineffectiveness [19], role-related stress [20], lack of rewards [21], and insufficient social support [22]. Consequently, burnout is associated with negative effects such as job dissatisfaction [23].

In the case of social workers, factors that predict burnout include social, demographic, and political transformations that affect service delivery [24]; barriers to professional development, such as low salaries, reduced social status, and limited resources for work [25,26]; stress related to conflict and role ambiguity [27]; and lack of support in the work environment and high turnover rates [28].

This burnout leads to various negative consequences for social workers, such as depression [29], intention to quit [30], self-reported health problems [31], and job dissatisfaction [32]. In addition, some studies highlight that work–family conflict (WFC) acts as an antecedent of burnout. For example, Ref. [33] evidenced the impact of WFC on burnout and the well-being of workers in the child welfare sector. However, more research is still needed to understand the structural mechanisms linking job support, WFC, burnout, and job satisfaction.

When an individual faces difficulties in managing a stressor, a maladjustment may occur that manifests itself psychologically (fear and anxiety) [34] or physiologically, affecting systems such as the digestive system, blood pressure, or immune response [35,36,37]. This maladjustment may compromise the physical functioning of the worker [38,39].

Emotional exhaustion, a central component of job burnout, describes a psychological state of work-related physical and emotional fatigue [40]. Stressors in the work environment, such as aggression, can drain an employee’s emotional resources, leading to this exhaustion [41,42]. When a worker faces aggression in their work environment, their personal resources are diminished due to constant worry, which intensifies emotional exhaustion. This exhaustion has significant negative implications, including a reduction in job satisfaction, organizational commitment, and job performance [43,44].

Previous research has confirmed the relationship between workplace aggression and emotional exhaustion, as well as the impact of victimization on emotional exhaustion [45,46,47]. However, there is little evidence on how aggressive behavior, i.e., being the aggressor, influences employees’ emotional exhaustion.

Studies examining the relationship between mentally strenuous work and burnout have yielded consistent results across the board: emotionally charged occupations are, by themselves, a predictor of the risk of burnout and the occurrence of some of its main components [48,49,50]. Furthermore, social workers engaged in field work have been identified as facing a particularly high risk. Compared to other social workers, they are more likely to experience physical and mental burnout in both their work and personal lives [51].

Among the main factors that contribute to burnout, in addition to individual or personal characteristics (such as certain predisposing personality traits), the literature highlights the specificities of the work and social environment. Research carried out by [7,40] on health sector workers has consistently shown that factors related to the work environment are determining factors in the development of burnout, although personal characteristics also play a relevant role. According to Maslach’s model, this syndrome originates due to imbalances in at least one of the following work areas: 1. workload: excessive demands, long hours, or poorly defined tasks. 2. Control: lack of autonomy over assigned tasks or time management. 3. Rewards: insufficient remuneration or lack of recognition. 4. Work community: poor interpersonal relationships or little social support. 5. Justice: perceptions of inequality in workload or remuneration. 6. Shared values: incompatibility between the employee’s values and those of the organization.

Over the past four decades, burnout research has highlighted the relevance of organizational resources in buffering the impact of work-related factors, underlining the importance of job satisfaction as a crucial factor. A meta-analysis by [52] concluded that emotional exhaustion, the initial stage of burnout, is closely linked to variables such as job autonomy, participation in decision making, workload, and working hours.

The job demands–resources (JD-R) model, proposed by [10], provides a framework for analyzing burnout and engagement in this area, positing that job demands and available resources act as triggers for two main processes: health decline and motivation [53]. Facing high job demands without adequate resources can lead to prolonged burnout [54]. Thus, the JD-R model is relevant for assessing the occupational health of social workers [55].

According to [56], Figure 1 illustrates the job demands–resources (JD-R) model. At the top, the factors associated with the health deterioration process are presented, while at the bottom, within ellipses, the elements that are part of the motivational process are highlighted. The arrows drawn in the model indicate the direction of the relationships between the different factors of both processes, while the signs reflect the nature of these relationships.

This model proposes that, when faced with work demands, workers may experience stress. If they fail to adequately manage these demands, stress can evolve into exhaustion, ultimately leading to burnout. Conversely, those who are able to respond to work demands, thanks to the resources available to them, initiate a motivational process that fosters commitment to and engagement with their work.

The search for quality in the provisioning of services and care by social services and efforts to care for and promote the emotional well-being of social work professionals become high-priority requirements for the future challenges of the discipline [57].

The objective of this study was to synthesize and analyze the results obtained in previous studies that examine the antecedents and consequences of emotional exhaustion, as one of the main dimensions of burnout in social work professionals. Specifically, this study seeks to address the following questions: What factors contribute to emotional exhaustion in professionals? How does emotional exhaustion affect job satisfaction and professional performance? What strategies can reduce the effects of emotional exhaustion? By exploring these questions, the study aims to provide a comprehensive understanding of emotional exhaustion in the field of social work.

## 2. Materials and Methods

### 2.1. Design

A systematic review was conducted in accordance with the recommendations and guidelines of the Joanna Briggs Institute [58]. To carry out this systematic research review, the Preferred Reporting Items for Systematic Reviews and Meta-Analyses (PRISMA Checklist) declaration was followed [59] Appendix A.

Figure 2 is shown below, following the PRISMA method. It provides a detailed visualization of this systematic review process, highlighting the number of studies included at each stage and the reasons for exclusion when applicable [23,24].

### 2.2. Search Strategy and Inclusion/Exclusion Criteria

The protocol being followed was registered in the International Prospective Register of Systematic Reviews (PROSPERO) with reference number CRD42024602943 prior to the identification and selection of studies. The search and selection process for the scientific articles that were the subject of this study was carried out between February and September 2024, obtaining a total of 361 records in five databases (Web of Science, Scopus, PubMed, MEDLINE, and PsycInfo). The search terms included the following equation: “Emotional Exhaustion” and “Social Work*”, using the Boolean operator (AND) in each database. No additional filters were applied, and the same search terms were used for articles written in a language other than English.

Restrictions: Studies published between 1 January 2019 and 30 September 2024 (last five years) were included, since our interest in the study is focused on ascertaining the most recent research (scientific articles) on this topic. Before the COVID-19 pandemic, social work professionals already presented high levels of emotional exhaustion [60,61] and, after its onset, an increase is to be expected, so much so that many social workers have indicated that “the main problem that Social Work faces after the pandemic is related to stress, emotional exhaustion and mental health” [62].

The study period on emotional exhaustion and burnout is limited to the years 2019–2024, primarily due to the significant impact of the COVID-19 pandemic. This period has seen increased stress and burnout among health and social work professionals, driven by unprecedented demands and the fear of contagion. Additionally, it allows for an analysis of how interventions and policies during and after the pandemic have influenced burnout prevalence and management, offering an updated and relevant perspective. The socioeconomic and political context, marked by austerity measures affecting the public sector and social services, has also increased pressure on social workers.

Inclusion criteria: (1) design criteria: cross-sectional studies; (2) language: English, Spanish, French, and Portuguese; (3) the sample of articles was social work professionals in professional practice; (4) the dimension of burnout of emotional exhaustion was addressed; (5) the studies were quantitative in methodology; and (6) the subject of the study addressed the antecedents and consequences of burnout.

The inclusion criteria are justified by the need to obtain a comprehensive and accurate understanding of burnout among social work professionals. Cross-sectional studies were chosen to identify correlations at a specific point in time, while including articles in English, Spanish, French, and Portuguese allows for a more global and diverse analysis. Focusing on practicing professionals ensures that the data are relevant and current. The emphasis on emotional exhaustion is due to its direct impact on work effectiveness. Quantitative methodology provides objective data and robust statistical analysis. Finally, addressing the antecedents and consequences of burnout allows for the identification of both underlying causes and long-term effects, facilitating the development of preventive and supportive interventions.

Exclusion criteria: (1) language criteria; (2) low scientific quality, according to the CRF-QS of [63]; (3) design criteria: studies published in conference proceedings, conference abstracts, and theses or studies that include animals were also excluded; (4) population: the sample of the studies was not practicing social work professionals; (5) Qualitative and mixed-methodology studies; and (6) any study that is not cross-sectional.

The exclusion criteria are justified to ensure the quality and relevance of the studies included in the analysis. Studies in unspecified languages were excluded to avoid interpretation errors. The exclusion of studies with low scientific quality, according to the CRF-QS by [63] Law et al., 1998, ensures that conclusions are based on rigorous and reliable research. Additionally, publications in conference proceedings, abstracts, and theses were excluded as they often do not undergo as rigorous a peer-review process as journal articles. Studies that do not include practicing social work professionals were excluded to ensure that the data are directly applicable and relevant to the current context. Quantitative studies were preferred over qualitative or mixed methods to obtain objective and statistically analyzable data. Finally, non-cross-sectional studies were excluded, as cross-sectional studies are ideal for assessing prevalence and correlations at a specific point in time.

## 3. Results

### 3.1. Selection of Studies

The search in the five databases yielded a total of 361 records, imported using the “Rayyan” web tool [64] to automatically detect and eliminate duplicate studies, resulting in the elimination of 230. Using the same tool, taking into account the titles and abstracts of the remaining 131 records, they were independently reviewed by two researchers, resulting in a total of 71 studies excluded for not meeting the inclusion criteria. In cases of conflict, a third researcher was consulted to make the final inclusion/exclusion decision. Then, once the 36 full-text articles were obtained, they were reviewed again independently by two of the researchers, excluding a total of 15 studies and leaving a total of 21 records. Figure 2 shows the process followed according to the PRISMA 2020 guidelines [59].

### 3.2. Evaluation of Methodological Quality

The reviewers independently established the methodological quality of the research selected through the critical evaluation instruments for CRF-QS studies of [65]. These tools allow for the evaluation of the methodological quality of a study and make it possible to determine to what extent a study has excluded or minimized the possibility of bias in its design, completion, and/or analysis. The following Table 1 details the level of methodological quality of the articles selected for the systematic review according to the score of items of 1 to 19, Appendix A complete.

This analysis has followed the studies [65,66,67,68,69,70,71,72,73,74,75,76,77,78,79,80,81,82,83,84,85,86,87,88]. As for the quality levels of the articles reviewed, they range from “poor” (score 10) to “very good” [89] (score 16), with an average quality of an “acceptable” level (average score 13.8) [90].

**Table 1 healthcare-13-00552-t001:** Methodological quality level according to the CRF-QS items [63].

Article	Total
[67]	15
[68]	16
[69]	12
[70]	15
[71]	15
[72]	13
[73]	16
[74]	16
[75]	12
[76]	16
[77]	14
[78]	15
[79]	16
[80]	15
[81]	15
[82]	14
[83]	16
[84]	13
[85]	13
[86]	14
[87]	16

Note: Maximum score of 19: poor quality level (≤11 points), acceptable (between 12 and 13 points), good (14 and 15 points), very good (16 and 17 points), and excellent (18 and 19 points).

With respect to the most repeated limitations of the quality analysis of the systematic review articles, Item 6, on the justification of the sample size, was not met in seven publications. Items 11 and 12, regarding the avoidance of contamination and cointervention, are evident in 21 articles, as is Item 15 on the presence of job absenteeism, which is present in 20 articles. Items 18 and 19, on reports of the clinical implication of the results obtained and descriptions of the limitations of the study, are reflected in eight and seven publications, respectively [63].

### 3.3. Analysis of Selected Studies

A total of 21 studies published from 1 January 2019 to 30 September 2024 were included in this study. Table 2 shows the main objectives, the methodology used, the data collection and scales, the type of analysis, and the most important results of the articles included in the present systematic review.

### 3.4. Study Design

The research included in this analysis follows a cross-sectional design, characterized by the collection of data at a single point in time. This method is common in studies on emotional exhaustion, as it facilitates analysis of the current state of essential factors, such as stress, social support, and job burnout, in large population groups. In addition, this design allows for the identification of connections between predictive factors, such as perceived stress or disputes at work, and their effects, such as emotional exhaustion or job satisfaction. However, an intrinsic restriction of this type of design is the inability to determine causal links, since variations over time are not analyzed, nor are longitudinal dynamics taken into account.

### 3.5. Study Population and Sample Size

The studies reviewed cover social workers working in a variety of settings, including health, community services, and government or private organizations, spanning a range of ranks, from frontline staff to supervisors and managers. Sample sizes fluctuate considerably, from 121 participants in Maddock’s 2023 [79] study in Northern Ireland to large studies with over 5000 social workers in China, such as those conducted by [78,84]. In summary, the data cover over 24,000 social workers, with sample sizes varying between 121 and 5965 individuals.

The geographical scope of the studies covers diverse regions of the planet. In Europe, multiple studies were carried out in countries such as Spain, Portugal, France, Germany, and Hungary, evidencing a strong interest in researching the dynamics of burnout in European work environments. In Asia, studies carried out in China, South Korea, and Israel highlight the influence of the sociocultural environment on the expression of emotional exhaustion. Despite their lower representation, research in the Middle East and Latin America, particularly in Saudi Arabia and Brazil, shows unique work challenges in these environments, such as social and economic stress, which significantly impact the mental health and well-being of social workers.

### 3.6. Measuring Instruments and Methods

The studies reviewed used a variety of techniques and procedures to measure burnout and its related elements. The Maslach Burnout Inventory (MBI) was the most widely used scale and was culturally adapted in multiple nations to assess three key dimensions of burnout: emotional exhaustion (EE), depersonalization (DP), and life satisfaction. In most research, the reliability of the MBI, especially for EE, was high, with Cronbach’s alpha values exceeding 0.80. In addition, the Perceived Stress Scale (PSS) was widely used to measure the general level of stress suffered by participants. Other individual surveys incorporated culturally validated instruments to assess social support, as well as scales developed to measure resilience, job satisfaction, and work–family conflict. In addition, mental health instruments such as the Hospital Anxiety and Depression Scale were used to assess the effect of stress and burnout on anxiety and depression.

Regarding statistical analysis, various methods were used. Descriptive assessments made it possible to describe the samples through frequencies and averages. Correlation studies, using Pearson and Spearman coefficients, facilitated the detection of direct links between key variables. Regression models, either linear or logistic, were used to establish relevant predictors of EE and examine connections with binary outcomes, such as the existence or not of EE. Additionally, mediation and moderation research, such as that conducted by [78,84], used sophisticated macros, such as PROCESS, to examine the interrelations between contextual and personal elements that influence EE.

Finally, in studies such as those by [71,87], structural equation models (SEMs) were used to understand complex links between various variables and validate theoretical models of EE, offering a more complete perspective of the elements that contribute to EE in different work environments.

### 3.7. Main Descriptive Results

The studies reviewed identified several risk factors, protective factors, and impacts related to emotional burnout among social workers. Among the risk factors, workload stood out as a component notably linked to emotional burnout in several studies, such as those conducted by [70,72]. Likewise, work–family disputes were presented as strong predictors, particularly in Asian settings, as reported by [77,84]. Other essential elements are workplace harassment and aggression, which were found to be strongly linked to emotional burnout in research conducted in Israel and France [74,81]. Finally, a lack of social support was a commonly identified predictor in almost all studies, particularly in the social services and health fields.

Regarding protective factors, resilience and self-care were found to be effective in reducing stress and emotional fatigue, with a particularly significant effect during the COVID-19 pandemic, according to [72,80]. Perceived social support, whether from peers or supervisors, also emerged as an essential shield against EE, as demonstrated in the research of [78,81].

Regarding the effects of EE, one of the most common impacts was a considerable decrease in job satisfaction, detected in employees with high levels of EE in research such as that by [69]. Additionally, a positive relationship was detected between EE and turnover intention, that is, willingness to leave one’s job, as indicated in the research by [84,85]. Finally, a correlation was found between EE and higher levels of anxiety and depression, negatively impacting the mental well-being of social workers, as recorded by [79,84]. These findings underline the importance of implementing strategies that address both risks and protective factors to mitigate the adverse effects of EE.

In the context of the reviewed studies, several key results related to burnout syndrome and turnover intention among social workers have been identified. [69] address the prevalence and risk factors associated with burnout syndrome among social work professionals in Spain, although effect sizes are not specified in the provided context. On the other hand, ref. [70] mention a correlational analysis of the dimensions of the MBI with Cronbach’s alpha values, indicating internal reliability, but specific effect sizes are not provided.

Regarding other articles, the provided contexts do not include specific details about effect sizes or standardized comparisons. To obtain this information, it would be necessary to access the full articles and review the results sections where these measures are generally reported. It is recommended to look in the results or discussion sections where effect sizes, such as Cohen’s d, odds ratios, or standardized mean differences, are usually presented.

In the study by [76], several standardized effects related to stress and burnout are presented. For example, the total effect of stress on the Maslach Burnout Inventory-Personal Accomplishment (MBI-PA) is −0.374, with a 95% confidence interval between -0.443 and -0.297, and is significant with *p* < 0.001. The indirect effect of stress on the MBI-PA is -0.239, also significant with *p* = 0.001. In the article by [70], a correlational analysis of the dimensions of the MBI with Cronbach’s alpha values is conducted. Significant correlations include EE with personal accomplishment (PR) with r = 0.328 ** and with depersonalization (D) with r = 0.381, both significant at *p* < 0.001.

Other studies do not provide specific details about effect sizes or standardized comparisons in the provided contexts. However, it is mentioned that the results of some studies may have weak effect sizes due to the complexity of stress and burnout mechanisms. These effect sizes and standardized comparisons are useful for understanding the magnitude and direction of the relationships studied in the mentioned articles.

In the provided contexts, some effect sizes and specific results related to burnout and job satisfaction among social workers are mentioned. In one study, several effect sizes related to burnout and job satisfaction are reported. For example, burnout–personal accomplishment (PA) positively predicts job satisfaction with an effect size of β = 0.24 (*p* < 0.00), while burnout–EE negatively predicts it with β = −0.13 (*p* < 0.00). Additionally, a small indirect effect of burnout–PA mediating between work support and job satisfaction is mentioned with an effect size of 0.02 (95% CI = 0.00~0.04, *p* < 0.01).

In another study, a SEM is used to evaluate the effects of the interaction between government work and clients on social workers’ burnout, mediated by role conflict. An effect size of β = 0.020 (*p* < 0.01) is reported for role conflict as a mediator. These results provide a comprehensive view of how different factors and strategies can influence burnout and job satisfaction, highlighting the importance of considering both direct and indirect effects in the analysis of these phenomena.

## 4. Discussion

This analysis identified a total of 21 studies that examined the relationship between EE and different contextual variables. This relationship has been the subject of study in numerous international studies, demonstrating how work, personal, social, and economic elements are intertwined and favor the emergence of this syndrome in social work professionals.

One of the main causes of emotional burnout is workload. Several studies, such as the one carried out by [70] in Spain, agree that work intensity and lack of rest are crucial elements for the development of EE. These findings are replicated in studies from European and Asian settings, such as those carried out by [72,86], which also emphasize pressure at work as one of the key factors of burnout. Overwork, particularly when demands exceed available resources, contributes significantly to emotional fatigue and mental exhaustion, a trend detected in various geographical regions.

Another significant element in emotional burnout is the conflict between work and family obligations, known as WFC. Research conducted in China, such as that by [77,84], shows that this conflict not only increases emotional fatigue but also indirectly affects job satisfaction. The relationship between job demands and family seems to aggravate EE, indicating that work–life balance is essential to prevent emotional burnout. This pattern is consistent with research conducted in other contexts, where social workers face an extra burden due to having to balance multiple roles.

Social support is presented as an essential protective element in reducing EE. Studies carried out by [72], in France, and [78], in China, emphasize that a positive work environment, together with the support of supervisors and colleagues, is directly linked to lower levels of emotional fatigue. This type of support assists employees in managing stress, generating an environment where risk factors are reduced. However, the lack of social support, together with circumstances of harassment at work or victimization, causes the opposite effect, intensifying EE, as highlighted in the research by [74] in Israel.

In settings featuring socioeconomic challenges, emotional burnout is often more pronounced. In places such as [67] and Brazil, social workers suffer higher levels of EE, affected by economic and social stress. These findings indicate that structural inequities in the work environment, coupled with circumstances of economic vulnerability, are elements that intensify burnout. Therefore, improving working conditions and addressing socioeconomic disparities could significantly reduce the risk of burnout in these settings.

In addition to organizational and social tactics, there are personal elements that contribute to protecting against emotional burnout. Resilience and self-care have proven to be essential in preventing EE. Research in Europe, such as that conducted by [76], in Slovakia, indicates that self-care, coupled with job satisfaction, can mitigate the adverse impacts of stress. This highlights the relevance of promoting well-being programs aimed at social workers. In addition, work flexibility and remote work have been recognized, particularly during the COVID-19 pandemic, as protective elements against emotional burnout. Although teleworking offers certain benefits, such as flexibility, its impact on work–life balance may be limited if work expectations are not properly managed.

The correlation between emotional fatigue and contextual factors fluctuates significantly across different geographic areas. In Europe, research often focuses on workload and stress at work, highlighting the pressure on health sectors, as evidenced by [72]. In Asia, meanwhile, research shows that cultural conflicts and work–family balance are dominant elements. In nations such as China and South Korea these elements directly impact emotional fatigue and the desire to change jobs. In Latin America and the Middle East economic and social elements play a crucial role, intensifying job burnout in situations of high vulnerability, which underlines the importance of addressing structural inequalities in the workplace.

### Limitations and Future Research

Research on EE in social workers has significant limitations. Most studies employ a cross-sectional design, which precludes establishing causal links between variables. In addition, methodological diversity and cultural adaptations of the MBI make it difficult to compare studies. The lack of longitudinal studies also limits understanding of how EE varies over time or with interventions.

It is recommended to implement wellness and resilience programs that promote self-care and organizational support, such as supervision and mentoring, to reduce the risk of burnout. It is also necessary to promote work flexibility, such as teleworking or adaptable schedules, to improve work–life balance.

Interventions to reduce stress and emotional exhaustion in social workers include support programs based on the clinically modified Buddhist psychological model, which focus on improving areas such as mindfulness, acceptance, attention regulation, self-compassion, detachment, and non-aversion. These programs have proven effective in reducing stress, emotional exhaustion, and depersonalization while enhancing personal achievement. Additionally, social support from coworkers and supervisors is crucial for mitigating emotional exhaustion, although supervision alone has been found insufficient to counteract burnout. Organizational interventions are also essential, focusing on reducing workload and emotional demands while increasing work resources such as autonomy and continuous support. Finally, promoting active coping strategies and resilience can help social workers better manage stress and improve their overall well-being.

This paper reviews studies on EE and stress among social workers in various contexts, such as China, Spain, and other countries, highlighting factors like workload, social support, and working conditions. It emphasizes the importance of improving social support in the workplace to mitigate burnout and workplace bullying, especially in high-demand and resource-limited settings. Policies should focus on reducing excessive work demands and enhancing job autonomy to prevent burnout, as suggested by studies from Spain and Germany. Training in emotional competencies and resilience is recommended to prevent burnout, particularly in economic crises and austerity policies. Future studies should evaluate the effectiveness of coping strategies like mindfulness and emotional regulation, investigate psychological protective and risk factors, and develop theoretical models to better understand the interaction between work demands and personal resources in EE. Additionally, the content suggests that fostering a good organizational climate of cooperation and trust, encouraging teamwork, and strengthening professional identity can reduce turnover intention and burnout among social workers.

## 5. Conclusions

The review of emotional burnout in social workers reveals that this is a global problem influenced by organizational, personal, and cultural factors. The main predictors of burnout include excessive workload, work–family conflicts, lack of social support, and unfavorable socioeconomic conditions. However, factors such as resilience, self-care, and organizational support can mitigate these effects.

To address the problem, it is recommended to develop wellness programs that encourage self-care, implement organizational support policies, and promote flexible work modalities to improve work–life balance. In addition, it is crucial to adapt strategies to the particularities of each region, considering work and cultural contexts to effectively prevent EE.

## Figures and Tables

**Figure 1 healthcare-13-00552-f001:**
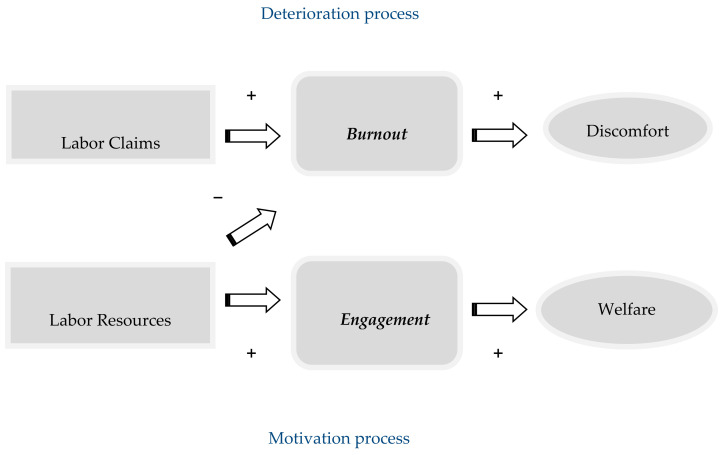
Labor Demand–Resource Model.

**Figure 2 healthcare-13-00552-f002:**
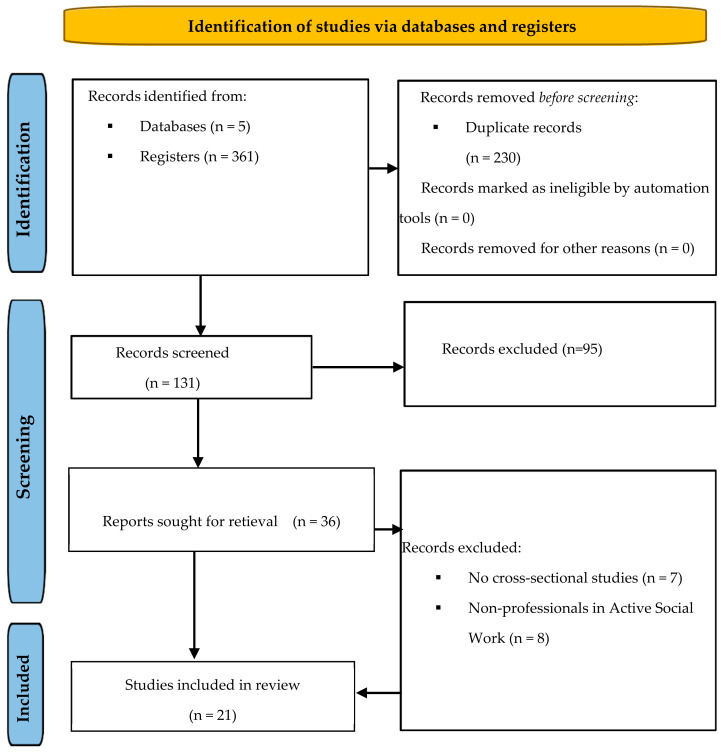
Bibliographic search flow chart.

**Table 2 healthcare-13-00552-t002:** Characteristics of articles included in this systematic review.

Authors	Design	Participants	Aim(s)	Data Collection and Scales	Analysis	Main Results
[67]	Cross-sectional study.	135 social workers from 22 social services centers for people with disabilities, divided into public, private, and NGO sectors.	Understanding the impact of social, family, and economic stress on the degree of emotional exhaustion experienced by workers who work with people with disabilities.	Design of a data collection tool (validated by experts). Using a modified version of the MBI-HSS scale to measure burnout, with a Cronbach’s alpha of 0.830 for EE, 0.790.	Using descriptive statistics for demographic data, Pearson correlation coefficients, *t*-test, ANOVA, and simple and multiple linear regression to predict R2.	Family stress associated with social stress is the variable that has the greatest capacity to predict the EE faced by social workers.
[68]	Cross-sectional study.	170 social workers from 142 residential care social services for adults with disabilities.	To study and analyze the influence of burnout syndrome on the job attractiveness of social workers.	Diagnostic method by K. Maslach, S. Jackson adapted from NE Vodopyanova. Application of the method for studying work-related burnout syndrome—Maslach Burnout Inventory (MBI).	Descriptive statistics, chi-square test, Kolmogorov–Smirnov normality test, Student’s *t*-test, Spearman’s non-uniform correlation metric, ANOVA.	High emotional exhaustion in social workers is associated with negative self-evaluation, alienation, irritation, and intolerance, affecting interpersonal relationships.
[69]	Cross-sectional study.	The sample of the study consisted of a total of 252 social workers of the Colleges Social Work Professionals in Seville and Murcia.	Analyze the prevalence of burnout and factors of risks associated with demographics, occupational, perceived social support, anxiety, and job satisfaction in a group of social workers in Spain.	Perceived social support (Duke-UNC), with Spanish version validated. Generalized Anxiety Disorder Scale. Job Satisfaction Questionnaire. Maslach Burnout Inventory.	Correlational analysis. Ordinary logistic regression analysis.	Burnout syndrome is significantly associated with higher anxiety and lower job satisfaction. The global dimension of burnout is statistically related to higher anxiety and lower job satisfaction.
[70]	Cross-sectional study.	947 social workers belonging to 36 Spanish professional associations.	To estimate the prevalence of burnout syndrome in a sample of Spanish social workers. To analyze the influence of a series of sociodemographic variables that may potentially be related to the appearance and development of one of the three dimensions of burnout.	Self-administered demographic questionnaire. Spanish version of the Seisdedos MBI (α = 0.90) for EE.	Correlation coefficients between variables were calculated, and hierarchical linear regression analysis was used.	For emotional exhaustion, the most important predictive variables were having been on sick leave, suffering from a chronic illness, having children, and being younger and inexperienced.
[71]	Cross-sectional study.	947 social workers belonging to 36 Spanish professional associations.	To analyze the relationships between the dimensions of burnout (EE, DP, and PA) linked to the theoretical model proposed. To analyze the relationships between burnout dimensions (EE, DP, and PA) and job satisfaction (JS).	Maslach Burnout Inventory, adapted for Spain: high reliability in EE (α = 0.89). Job satisfaction survey (α = 0.90).	Structural equation modeling using partial least squares modeling approach (PLS-SEM).	Social workers who experience a sense of EE in relation to their work and the working relationships established within the organization may exhibit reduced levels of satisfaction. EE is determined by workloads affecting mental well-being.
[72]	Cross-sectional study.	336 social workers from Portugal (n = 252) and Spain (n = 84) in the health sector.	Understanding the impact of COVID-19 on social workers working in the healthcare field, analyzing their levels of resilience and burnout.	Self-administered online questionnaire. Maslach Burnout Inventory-Human and Services Survey (MBI-HSS) (α = 0.762). Adaptation into Portuguese and Spanish of the Brief Resilient Coping Scale (BRCS).	Quantitative, descriptive, and inferential analysis, verifying the assumptions for parametric tests with Kolmogorov–Smirnov, ANOVA, and K-means cluster.	Social workers in services for patients with COVID-19 have a higher frequency of EE. Primary care social workers present the highest levels and prevalence of EE. Perceived support within the healthcare unit is a protective factor.
[73]	Cross-sectional study.	261 social professionals.	To examine how characteristics of the workplace and work organization influence burnout. Explore how working conditions, client conflicts, and cultural challenges affect the severity of burnout symptoms. Present how additional factors such as field of work and experience influence burnout.	Emotional exhaustion: Maslach–Bournot Inventory-HSS specialized for Hungarian workers, with the consent of Mind Garden (α = 0.7). Own measurement tool based on the models: Requirement–control–social support. Effort–reward inequality model. Requirements–resources model.	Factor analysis and multinomial logistic regression.	Work and organizational conditions and difficulties with clients are the factors that have the greatest effect on EE.
[74]	Cross-sectional study.	470 social workers from social services departments throughout Israel.	To examine the effects of aggression and victimization on psychosomatic stress symptoms.	Emotional exhaustion: Maslach Burnout Inventory (MBI) (α = 74). Stress symptoms: subscale of the Hopkins symptom checklist (Cronbach α = 84). Service Climate: Service Climate Scale, α = 0.91 and α = 0.88 in the two samples.	Linear regression models.	Aggression and victimization were the most significant predictors of EE.
[75]	Cross-sectional study.	341 active social workers.	To describe and empirically analyze the professional activities of social workers with specific job demands and working conditions in Germany.	BIBB/BAuA Labour Force Employment Survey on Qualifications and Working Conditions in Germany 2018. The population was surveyed using computer-assisted telephone interviewing (CATI).	Descriptive statistics. Correlation analysis.	The conditions of work intensity (speed of work, simultaneous work, work capacity, and work disturbances) determine the ability to feel EE.
[76]	Cross-sectional study.	689 social workers from 14 institutions in 8 districts of Slovakia.	To explore the link between perceived stress and the three dimensions of burnout (emotional exhaustion, depersonalization, and personal accomplishment), while considering the potential mediating role of self-care and job satisfaction that may prevent the transformation of stress into burnout.	Maslach Burnout Inventory-Human Services Survey (MBI-HSS). Perceived Stress Scale (PSS). Self-care activities questionnaire (VSS) Job Satisfaction Survey (JSS)	Ordinary least squares regression analysis.	The relationship between stress and emotional exhaustion is partially mediated by job satisfaction in terms of salary, nature of work, and operating procedures. All of these factors, along with self-care, are related to EE and may buffer its development.
[77]	Cross-sectional study.	1414 social workers from nine cities in Guangdong Province.	To understand the structural process of job support, WFC, burnout, and job satisfaction among Chinese social workers.	L1G job satisfaction. Job Support Scale: senior manager (α = 0.89), direct manager (α = 0.92), supervisor (α = 0.90), and peers (α = 0.90). Chinese Work–Family Balance Scale with subscales of work–family conflict (α = 0.82) and family–work conflict (α = 0.86). Maslach Burnout Inventory, emotional exhaustion (α = 0.91).	Structural equation models.	WFC directly exacerbates all three dimensions of burnout and indirectly impairs job satisfaction through the partial mediation of emotional exhaustion and reduced personal accomplishment, role strain, and other demographic factors. Job support alleviated burnout and promoted job satisfaction.
[78]	Cross-sectional study.	5965 social workers from 979 social work agencies.	To examine a moderated mediation model with the main effect of person–job fit and depression, the mediating role of emotional exhaustion, and the moderating role of work.	Person–job fit: five-item scale (CSWLS 2019), validated for Chinese social workers, with a Cronbach’s α of 0.844. Emotional exhaustion subscale of the Chinese version of the MBI with an α of 0.912. Social Support Scale: support from supervisor (α = 0.922), colleagues (α = 0.907), and manager (α = 0.909). CES-D Depression Scale, with an α of 0.929.	Descriptive and correlational statistics, regression using Hayes (2013) Macro PROCESS (Model 4), bias-corrected bootstrapping based on 5000 samples, and moderated mediation analysis with PROCESS (Model 14).	The effect of a lack of this supervisor, coworker, and manager support is confirmed to be significantly correlated with emotional exhaustion and depression. Similarly, it was confirmed that emotional exhaustion mediated the effects of person–work adjustment on depression.
[79]	Cross-sectional study.	121 social workers.	Provide a greater understanding of the possible relationships between stress, burnout (emotional exhaustion, depersonalization, and personal fulfillment), anxiety, depression, and well-being in social workers.	Perceived Stress Scale (PSS) (α = 0.87). Maslach Burnout Inventory (MBI) (α = 0.75). Hospital Anxiety and Depression Scale (HADS) (α = 0.75 and α = 0.83 for the HADS-D and the HADS-A). Warwick–Edinburgh Mental Wellbeing Scale (Wemwbs) (α = 0.89).	Correlation analysis. Multiple and bivariate regression.	Perceived stress and emotional exhaustion significantly predicted anxiety. Emotional exhaustion did not significantly predict either depression or mental well-being.
[80]	Cross-sectional study.	121 social workers.	Provide a greater understanding of what the possible outcomes might be. Risk and psychological protection factors for stress and depression burnout of social workers. Provide empirical evidence on whether: (a) a CBPM model of direct and mediated effects is directly associated with stress, exhaustion emotional, depersonalization, and personal fulfillment.	The Perceived Stress Scale (PSS) (α = 0.87). The Maslach Burnout Inventory (MBI) (α = 0.81). The Southampton Mindfulness Questionnaire (SMQ) (α = 0.74 and α = 0.6). The Experiences Questionnaire: Attention Regulation/Decentering (EQ-D) (α = 0.88). Philadelphia Acceptance and Mindfulness Subscale (PHLMS-A) (α = 0.88). The Self-Compassion Scale: Short Version (SCS-SF) (α = 0.86). Penn State Worry Questionnaire (PSWQ) (α = 0.94). Rumination Reflection Questionnaire: Rumination subscale (RRQ) (α = 0.93).	Structural equation models. Analysis of conditional processes.	It was shown, preliminarily, that a mediated-effect CBPM model can be formed as a potentially useful explanatory framework to analyze variations in emotional exhaustion. Preliminary evidence also showed that if social workers participate in support programs they are likely to enjoy reductions in emotional exhaustion.
[81]	Cross-sectional study.	369 social workers providing services in different social services centers.	To examine the mediating role of workplace bullying in the relationship between social support (family and friends; coworkers and supervisors) and emotional exhaustion.	Maslach Burnout Inventory-Human Survey (MBI-HSS) (α = 0.92). The scale of social support from family and friends (α = 0.92). The Coworker and Supervisor Social Support Scale (α = 0.87). The Belgian version of the Negative Acts Questionnaire (α = 0.91).	Structural equation models. One-way analysis of variance (ANOVA).	Social support from family, friends, coworkers, and supervisors was shown to contribute in different ways to reducing emotional burnout in social workers. Workplace bullying is strongly and positively associated with EE.
[82]	Cross-sectional study.	273 social workers from social services.	To determine burnout levels among social workers in Spain during the first wave of the pandemic and predictive variables.	MBI. The Cronbach alpha in the original version obtained an EE dimension of α = 0.89. In our version, adapted to Spanish, α = 0.90 for the EE dimension.	Descriptive and cross-analysis. Binary logistic regressions.	Teleworking and psychological treatment are predictive variables of emotional exhaustion.
[83]	Cross-sectional study.	316 social workers from social services.	To examine the mediating roles of emotional exhaustion and depersonalization.	Emotional exhaustion: Maslach Burnout Inventory Rotation intention: scale developed. Job demands: Korean Occupational Stress Scale-Short Form (KOSS-SF). Job resources: three-item social support subscale of KOSS-SF.	Bivariate correlation analysis, serial multiple mediation, and use of PROCESS, based on ordinary least squares regression and bootstrap method.	Job demands and turnover intention are significantly correlated with emotional exhaustion, whereas job resources are negatively related to emotional exhaustion. Furthermore, emotional exhaustion leads to depersonalization, both in contexts of high job demands and in situations with limited job resources.
[84]	Cross-sectional study.	5478 workers social services from 813 social service organizations in China.	To investigate the association between collective psychological ownership (CPO) and social workers’ turnover intention by controlling for psychological demands (emotional exhaustion and psychological resources).	Turnover intention: 4 items, α = 0.73. Role ambiguity: 5 items, α = 0.81. Emotional exhaustion: 9 items (Chinese version of the MBI-HSS) with α = 0.91. Autonomy: 2 items (Job Content Questionnaire, α = 0.75. Self-perceived POVC: 2 items (CSWLS expert panel 2019), α = 0.81. Self-perceived social support: Social Support Scale, α = 0.95. Self-perceived collective psychological ownership: 7 items, α = 0.82.	Multilevel regression analysis.	Social workers showed negative associations with gender, age, and job position and with all the psychological resources assessed. However, they were positively associated with educational level and emotional exhaustion. Furthermore, emotional exhaustion is positively related to turnover intention, while role ambiguity is not a significant predictor.
[85]	Cross-sectional study.	209 social workers from the social services of Galicia participated.	Evaluate the prevalence of burnout syndrome in female social workers who work in community social services.	The Spanish version of the Maslach Burnout Inventory questionnaire, Human Services Survey (MBI-HSS) was used (α = 0.87).	Descriptive analysis.	A high level of burnout has been confirmed among social workers in Galicia, with emotional exhaustion being the most prominent dimension. The main predictors are the increase in the number of demands and the high workload.
[86]	Cross-sectional study.	2643 frontline social workers and 2599 supervisors or managers.	To investigate whether tension between working with governments and clients might affect social workers’ burnout through the mediating effects of role ambiguity and conflict.	Emotional exhaustion: Chinese version of the MBI-HSS, using a Likert scale. Cronbach’s alphas for the three subscales were 0.918, 0.816, and 0.921. Role stress was measured with the Chinese version of the Role Ambiguity and Conflict Scales.	Structural equations.	Social workers in China who interact more with government departments and less with clients may experience greater role conflict, which could increase their burnout. However, this is not observed for workers with greater management or supervisory burdens.
[87]	Cross-sectional study.	5620 social workers.	To examine how workplace social capital affects turnover intention in public service.	Workplace social capital was measured with a validated Chinese version of an eight-item scale (α = 0.916–0.927). Emotional exhaustion: it was measured with the Chinese version of the MBI-GS (α = 0.893–0.922). Job satisfaction: was assessed with a validated Chinese version with four items (α = 0.898).	Confirmatory factor analysis. Regression analysis. PROCESS 2.16 developed by Hayes (2013).	Emotional exhaustion and job satisfaction mediated the relationship between social capital at work and turnover intention. Public service agencies should foster a climate of cooperation and trust and promote teamwork and altruistic behaviors to reduce emotional exhaustion and strengthen both professional identity and the value of social work.

Source: Prepared by the authors based on the results obtained in the systematic review (2024).

## Data Availability

Data are available on request from the authors.

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
