# Peer review of "Analysis of the Predictors and Consequential Factors of Emotional Exhaustion Among Social Workers: A Systematic Review"

_healthcare, 2025, doi:10.3390/healthcare13050552_

Round 1
Reviewer 1 Report
Comments and Suggestions for Authors
Dear authors, first of all, congratulations on your review work. Below are the suggestions that need to be made to improve your study:
1. In the introduction, it is necessary to adapt the way of citing references based on the journal. That is, there is no space between number and number of references. In relation to the theoretical foundation, it is well founded.
2. The method is appropriate, they have followed the steps proposed by the PRISMA statement and have assessed the risk of bias. Excellent!
3. In the results section, in the table it is necessary to put the citations in the first column (author) in Vancouver format, as established by the magazine. On the other hand, the appointments have to be changed from APA 7 to Vancouver. Otherwise, everything is very well described and detailed.
4. The discussion is also correct, in it they discuss their results with previous literature. However, it is necessary to change the appointments to Vancouver. Furthermore, they must point out in the discussion that the results obtained are from a merely descriptive study.
Author Response
Comments Reviewer 1:
Thank you very much for your comments and for your dedication in reviewing the manuscript. The references have followed the guidelines and examples in the template provided by the journal and have been followed in full completeness.
1.Changed throughout the text citations and references to Vancouver.
3.Changed (line 278).
4.Changed and specified descriptive (line 333).
Reviewer 2 Report
Comments and Suggestions for Authors
I read with interest the paper titled "Analysis of the predictors and consequential factors of emotional exhaustion among social workers: a systematic review"
I have some comments to enhance the systematic review:
1. PRISMA is a way to report systematic reviews, not a guideline to perform them. Saying that the review was conducted based on PRISMA is a quite common mistake, since they have no guidelines for conduct this type of research, rather than how to present it. As PRISMA acronym says, those are the Preferred REPORTING Items for Systematic reviews and Meta-Analyses, not guidelines for conducting it.
2. In addition, the paper is not reported using the PRISMA guidelines, because, this means that authors use the PRISMA checklist and divide the paper in sections and topics that are reported in PRISMA checklist.
3. In addition, since authors state that they use the PRISMA for reporting, the checklist should be added as supplementary material.
4. Emotional exhausting and burnout are not new topics. What's the rational of the period of study being limited to 2019-2024 years.
5. Furthermore in the abstract, authors stated that the search was conducted between 2019-2024. In the PROSPERO the search was intended to be done between 1 January 2019 and 30 September 2024, meaning that 2024 is not a complete year, so month/year date is preferred.
6. According to PRISMA, quality assessment, should be presented as results, rather than in methods section.
7. I suggest that search to be added in the manuscript (as per PRISMA checklist, point 7)
8. Risk of bias should be performed in the systematic review.
9. The results section is missing the core content of a systematic review. For all outcomes, present, for each study: (a) summary statistics for each group and (b) an effect estimate and its precision (e.g. confidence/credible interval), ideally using structured tables or plots. Then you should move to data synthesis and briefly summarise the characteristics and risk of bias among contributing studies. Present results of statistical syntheses conducted, if any. Present results of all investigations of possible causes of heterogeneity among study results. Present results of all sensitivity analyses conducted to assess the robustness of the synthesized results: this information is almost missing in the paper.
Author Response
Comments Reviewer 2
Thank you very much for your review of the manuscript and for your time. All changes suggested by you have been marked in yellow and thank you for all your considerations and improvements to the manuscript.
- Changed (lines 16 and 17).
- Adapted in the text.
- Checklist has been included as supplementary material.
- Search strategy and inclusion/exclusion criterion (line 180).
- Changed (line 19).
- Changed (line 248).
- The search equation was already entered (line 186).
Reviewer 3 Report
Comments and Suggestions for Authors
The manuscript addresses the important and relevant issue of emotional exhaustion given the increasing prevalence of burnout in social work. The findings demonstrate factors, such as workload, work-family conflict, and social support, as contributors to burnout. It could emphasize unique contributions, such as regional disparities or emerging trends post-pandemic, on under-researched factors, such as organizational culture, technological influences, or longitudinal changes, would enhance its impact and the novelty.
The stated objective is clear. However, the research questions could be more explicitly defined in the introduction. The study aims to analyze antecedents and consequences, but the methodology does not sufficiently distinguish between these aspects in the results section.
The results section effectively presents key findings, but the quantitative synthesis is weak; meta-analytic techniques could strengthen conclusions. Effect sizes or standardized comparisons across studies are missing.
The manuscript specifies recommendations such as work flexibility, wellness programs, and organizational support. However, implications for policy/workplace interventions, future intervention studies could be explored further.
The manuscript is well-written, but some areas need improvement:
Line 112 and 118. Are “JD-R” and “JDR” the same model? If yes, please use the format consistently.
Line 125. Figure 1 “Labor Demand-Resource Model” does not correspond to the in-text citation of Figure 1 (line 239). Please add the in-text citation for Figure 1 close to the figure.
Line 176. "The search terms included the following equation: 'Emotional Exhaustion' and 'Social Work*' using the Boolean operator (AND) in each database." → Were any additional filters applied? The same search terms applied for articles written in other than English?
Line 243. The standard use of PRISMA and JBI guidelines are appropriate. Justification for the inclusion of only cross-sectional studies (Line 243~), and for the timeframe (2019-2024) appears weak (bias towards COVID-19). The difference in methodological quality among included studies could be discussed more critically (Table 1). It lacks in-text citation for Table 1.
Lines 256-257, 280, 289, 292 etc.: please use the required citation format.
Lines 272, 276, 281: add references for MBI, PSS etc.
Line 327. Abbreviations are inconsistently used (line 327, 384 etc. no need to mention full text “emotional exhaustion” for "EE" since it was introduced in the previous section. Similarly, for MBI in line 387.
Line 363. "Resilience and self-care were found to be effective in reducing stress and emotional fatigue..." → Specify which interventions are most effective.
Line 403. "To address the problem, it is recommended to develop wellness programs that encourage self-care...", "It is crucial to adapt strategies to the particularities of each region..." → Provide more concrete examples of effective programs to the Discussion section, and examples of region-specific interventions.
Lines 424-428: Abbreviations section does not include all abbreviations used. Please consider if this section is necessary.
Author Response
Comments Reviewer 3
Thank you very much for your review of the manuscript and for your time. All changes suggested by you have been marked in yellow and thank you for all your considerations and improvements to the manuscript.
- Research questions have been included (line 138).
- The statistics used in some articles have been included (line 358).
- This has been expanded in the discussion part (line 464).
Round 2
Reviewer 2 Report
Comments and Suggestions for Authors
The authors didn't provide an answer to all my concerns regarding the study flawlesses.
In the author's reply, they provide answer to 7 out of 9 points. No explanations were provided, just small sentences refering to parts of the text and ambiguous answers.
Furthermore, I don't understand why authors insist in the idea that they are using PRISMA for reporting the review, since they fulfilled the table with page numbers, but when checked the review, some topics are missing and are not found in the review. I understand the desire to refer PRISMA as a guideline for systematic reviews, but when the intention is to use, the review should be reported accordingly, and I feel that authors are just fulfilling the steps to get the paper accepted, but without care of what are doing.
When a review is stating that is using PRISMA for reporting, I expect to open the review and see the exact same topics that are present in the checklist, including all the sections, all the topics and all the items that the checklist present. It's ok to not use PRISMA, but then just don't add it as a cherry in your review. If the authors want to use it, so use it properly.
As an example, Certainty of evidence, reporting bias, and their evaluations are not even close of what is expected from a systematic review.
In this sense, my decision is to reject.
Author Response
Dear Reviewer:
I thank you for your comments, but respond below to your concerns.
In response to point 8, the risk of bias in the systematic review has been done.
In response to point 9, this point has been made and can be seen in the manuscript between line 358 and 397.
The rest of the improvements have been made in detail and in consideration of your comments and recommendations.
Finally, I disagree on the use of the PRISMA guide, I have read many, many studies that do not follow the checklist point by point and even in teacher training courses in relation to PRISMA and they have not said that the manuscript should answer the checklist point by point.
Again, thank you for your time and dedication to the review of the manuscript and I would appreciate your reconsideration of the decision, as a lot of effort and attention has been given to your recommendations, changes and improvements to the submitted manuscript.
Reviewer 3 Report
Comments and Suggestions for Authors
The authors responded adequately all of my comments, I think the manuscript is improved now. I have no further comments.
Author Response
Thank you very much for your time and dedication to the review. We are very grateful for your decision.